# A Rationale and Approach to the Development of Specific Treatments for HIV Associated Neurocognitive Impairment

**DOI:** 10.3390/microorganisms10112244

**Published:** 2022-11-12

**Authors:** Aaron Scanlan, Zhan Zhang, Rajeth Koneru, Monica Reece, Christina Gavegnano, Albert M. Anderson, William Tyor

**Affiliations:** 1Atlanta Veterans Affairs Medical Center, Decatur, GA 30033, USA; 2Department of Neurology, Emory University School of Medicine, Atlanta, GA 30322, USA; 3Department of Pathology, Division of Experimental Pathology, Emory University, Atlanta, GA 30322, USA; 4Department of Pharmacology and Chemical Biology, Emory University, Atlanta, GA 30322, USA; 5Department of Medicine, Division of Infectious Diseases, Emory University School of Medicine, Atlanta, GA 30322, USA

**Keywords:** HAND, HIV neurocognitive impairment, neuroHIV, HIV, brain, adjunctive therapy, neuroinflammation

## Abstract

Neurocognitive impairment (NCI) associated with HIV infection of the brain impacts a large proportion of people with HIV (PWH) regardless of antiretroviral therapy (ART). While the number of PWH and severe NCI has dropped considerably with the introduction of ART, the sole use of ART is not sufficient to prevent or arrest NCI in many PWH. As the HIV field continues to investigate cure strategies, adjunctive therapies are greatly needed. HIV imaging, cerebrospinal fluid, and pathological studies point to the presence of continual inflammation, and the presence of HIV RNA, DNA, and proteins in the brain despite ART. Clinical trials exploring potential adjunctive therapeutics for the treatment of HIV NCI over the last few decades have had limited success. Ideally, future research and development of novel compounds need to address both the HIV replication and neuroinflammation associated with HIV infection in the brain. Brain mononuclear phagocytes (MPs) are the primary instigators of inflammation and HIV protein expression; therefore, adjunctive treatments that act on MPs, such as immunomodulating agents, look promising. In this review, we will highlight recent developments of innovative therapies and discuss future approaches for HIV NCI treatment.

## 1. Introduction

Herein, we emphasize the need for adjunctive therapies for HIV infection of the central nervous system (CNS). As the HIV field continues to search for a cure, the adverse effects posed by HIV in the brain are an important consideration. Ongoing inflammation has been consistently demonstrated during HIV infection of the CNS, and is associated with cognitive dysfunction in people with HIV (PWH). We highlight potential adjunctive therapies (in addition to standard antiretroviral therapy) that are anti-HIV, anti-inflammatory, or both. In some cases, featured treatments have other beneficial effects for neuronal health, thus they are neuroprotective. We divided this review into the clinical aspects of HIV associated neurocognitive impairment (NCI), which includes discussion of the cognitive, imaging, cerebrospinal, HIV persistence, treatment, and pathological features relevant to PWH. The review then segues into research of adjunctive treatments in animal models and in vitro systems that can potentially be investigated in clinical trials. Finally, in the Future Directions for the HIV NCI treatment space we make recommendations regarding specific novel treatments to consider for human trials. While it is not feasible to cover every clinical trial or experimental compound for the treatment of HIV NCI, we endeavor to underscore treatments that address neuroinflammation or neuronal injury/damage linked to HIV infection in the brain.

## 2. Clinical and Neuropathological Features of HIV Associated Cognitive Dysfunction

### 2.1. Cognitive

Many clinicians and investigators who specialize in the neurological complications of HIV infection continue to use the Frascati criteria as a framework to categorize HIV associated NCI [1]. Three main categories of cognitive dysfunction within HIV associated neurocognitive disorders (HAND) are defined and include asymptomatic neurocognitive impairment (ANI), mild neurocognitive disorder (MND) and HIV associated dementia (HAD). ANI and MND are essentially defined by specific abnormalities in neuropsychological testing. The primary distinction is that PWH who have MND have mild functional deficits and those with ANI do not. Meanwhile, those with HAD have even more profound neurocognitive deficits than those with MND. Those with HAD have more significant impairment of activities of daily living (ADLs) than those with MND [2]. Clinical evidence indicates that ANI and MND can be reversed with initiation of antiretroviral therapy (ART), while HAD is often not completely reversible (see Section 2.5. Treatment). Therefore, ANI and MND can be considered ‘mild’ HAND or mild NCI. We focus on mild NCI in PWH because the primary emphasis of this review is to discuss the development of therapeutic strategies that reverse or stabilize mild NCI in PWH who are already on suppressive ART.

As a larger proportion of the PWH ages, risk factors for NCI and other medical comorbidities have a larger role in the PWH population with mild NCI [3,4,5]. In some cases, the presence of these comorbidities makes it difficult to discern whether the mild NCI is linked to HIV infection in the brain or another underlying condition. Comorbidities related to age may include hypertension, diabetes, cerebrovascular disease, or renal failure [3,4,5]. Another major comorbidity in the PWH population is the presence of traumatic brain injury (TBI) [6,7]. In a preliminary study of 41 veterans with HIV who were recruited sequentially at the Atlanta VA Medical Center, 44% had NCI and 20% of those had a history of TBI [8,9]. TBI appears to make cognitive impairment worse in PWH vs. HIV associated NCI without TBI as executive function and working memory are both significantly decreased [6]. Notably, it was found that approximately 1/5 of the participants in the CNS HIV Antiretroviral Therapy Effects Research (CHARTER) study group were reported to have TBI [7]. Other risk factors and comorbidities that are relevant to PWH with mild NCI also include ART-associated neurotoxicity, low CD4+ count, longer period of HIV infection, high systemic viral load, cardiovascular disease, alcohol and drug abuse, coinfection (e.g., syphilis), or other neurodegenerative diseases such as Alzheimer’s Disease and Parkinson’s Disease [10,11,12,13,14,15,16,17,18,19,20,21,22]. The list of possible comorbidities in PWH is extensive, which makes it currently difficult for the field to differentiate specific biomarkers linked to cognitive impairment in PWH.

### 2.2. Imaging

Autopsy studies (see Section 2.6. Pathology) provide invaluable information about the pathogenesis of HIV brain infection; however, they have limitations. These include alterations in pathophysiological findings related to various specific causes of death (e.g., systemic infections such as pneumonia), delays in times to autopsy that can result in spurious results, trouble with correlating mild NCI with pathology specifically related to it versus agonal effects in the dying brain, and relatively low number of cases coming to autopsy with mild NCI, making statistical considerations more challenging. Neuroimaging has the advantage of showing brain pathology without the above considerations and in live patients. This will be briefly reviewed herein, because there are extensive published reviews available [23,24]. Imaging has been used to study a wide variety of important features of the effects of HIV on the brain, including brain volume and neuroinflammation [23]. Drawbacks to current neuroimaging methods include the questionable ability to examine markers of neuroinflammation and neuronal injury and particularly the inability to detect subtle changes in finer structures like synapses or dendritic spines, which are critical features of mild NCI [24]. Magnetic resonance imaging (MRI) has been used to detect atrophy in areas such as the frontal and temporal cortex, which reflects neuronal loss and injury, in PWH [25]. These MRI studies lend support to the idea that ART is not sufficient in preventing white matter injury for some PWH [18,26]. 

Magnetic resonance spectroscopy (MRS) can be used in conjunction with MRI to examine the biochemical composition of tissue [27]. MRS measures such as N-acetylaspartate (NAA), choline (Cho), GABA, glutamate/glutamine complex (glx), Cr (creatine), and myo-inositol (Mi) have been used to characterize neuronal integrity and neuronal mitochondrial function, inflammation, neuronal receptors, metabolism, and glial activation, respectively [27,28,29,30]. MRS revealed that approximately 48% of HIV infected patients (neuro-asymptomatic and severe NCI) presented significant increases in inflammation and glial activation in the frontal white matter (FWM), basal ganglia (BG), and mid-frontal cortex (MFC) [31]. A decrease in neuronal integrity was observed in the severe NCI HIV-infected group [31]. As reiterated here and elsewhere in this review, conventional ART alone is not sufficient to prevent cognitive impairment and irregularities in brain metabolite levels and in addition, the long-term reduction in Glx and NAA in specific regions are associated with NCI [32]. This study suggests that despite HIV suppression, neuronal injury can still occur as indicated by the reduction in Glx and NAA in cortical and/or subcortical regions. 

Nevertheless, following ART, patients can exhibit a decrease in inflammation and glial activation in the FWM, parietal gray matter, and BG [33]. The study gives further support for early ART initiation in dampening neuroinflammation and inhibiting further neuronal injury [33]. Other studies using MRS underline the role of glial inflammation in cognitive dysfunction [34,35,36,37]. 

Other groups have used positron emission tomography (PET) technology to study potential markers of neuroinflammation in PWH [38,39,40,41,42]. TPSO PET studies in PWH suggest associations with specific cognitive domains and microglial activity in brain regions such as frontal cortex, hippocampus, anterior and posterior cingulate, and corpus callosum [39,40,43]. However, while TPSO is found to be highly elevated in microglia and astrocytes during neuronal injury, the biological specificity and relevance of TPSO to neuroinflammatory activity has been debated, underscoring that a multitude of measures may better provide utility towards clarity of information gained in the setting of PWH [42,44,45].

### 2.3. Cerebrospinal Findings

Aside from autopsy and neuroimaging studies, cerebrospinal fluid (CSF) investigations can provide insight into the pathogenesis of HIV brain infection. However, CSF at best only roughly reflects virological and immunological events occurring in brain parenchyma [46]. Interferon-alpha (IFNa) has been shown to be significantly elevated in PWH that have severe NCI [47,48,49]. Additionally, a cross-sectional study by Anderson and colleagues demonstrated a correlation between cognitive impairment and increased IFNa levels in PWH and mild NCI [50]. The same study also found a strong correlation between a rise in CSF neurofilament light chain protein and IFNa levels [50]. Williams and colleagues published a comparative review on CSF markers for HIV infection of the CNS [51]. Briefly, the authors screened 1943 studies and narrowed the final review to 20 cross-sectional studies and 9 longitudinal studies based on defined criteria. On closer examination, the authors found higher levels of CSF markers neopterin, sCD163, sCD14, IFN-γ, IL-1α, IL-7, IL-8, and sTNFR-II while lower levels of IL-6 are associated most consistently with NCI in PWH. The authors acknowledge the limitations of their comparative review due to: (1) age and sex-matching is not consistent across each study, (2) CNS/plasma discordance (higher viral load in CSF vs. plasma) not determined across all studies, (3) sampling bias mostly restricted to U.S. studies, (4) low number of participants leading to less statistical power, (5) most studies do not correlate CSF markers with disease severity/stage, and (6) confounding factors such as coinfections/comorbidities are not regularly reported in the 29 studies reviewed [51]. 

There is considerable evidence through CSF (see above) and autopsy studies (see below) that HIV persists within the brain when replication is suppressed in plasma by ART [52,53,54]. The importance of HIV persistence in the brain is predicated on two issues: (1) the potential for persistent brain HIV to re-seed the circulation and peripheral tissues, and (2) the capacity of persistent HIV in the CNS to affect the development and progression of HIV associated NCI. HIV persistence can be further defined as a tissue reservoir that allows for continual viral replication despite ART [55,56]. Criteria for a persistent viral reservoir can include: (1) confirmation of integrated proviral DNA in long-lived cells, and (2) continual presence of virus in a quiescent state in the reservoir that can be stimulated to generate replication competent virions [57,58]. Brain mononuclear phagocytes (MPs) such as perivascular macrophages can survive for months [59], while microglia can persist for years [60,61]. The number of brain MPs containing integrated HIV ranges widely depending on the neuroanatomic site of sampling, duration of HIV infection, ART selection and timing, and the presence or absence of NCI, which satisfies the first criteria above [62]. The presence of viral RNA in patient brains [63], and the detection of proviral DNA in brain MPs [62] suggests that the CNS represents another HIV reservoir that can support viral persistence in the era of ART, thus highlighting the need for adjunctive therapies.

### 2.4. Viral Persistence in CSF and Viral Escape

The presence of inflammatory cytokines and other biomarkers in the CSF mentioned above points to the persistence of HIV in the CSF. CNS seeding of HIV, as detected by HIV RNA in the CSF, can occur as early as a week after primary infection [64]. HIV proviral DNA can be detected in the CSF of PWH despite early administration of ART [65]. Furthermore, HIV RNA can be detected in the CSF up to 10 years on ART when testing is performed with assays that quantitate virus to very low levels (single copy assays), reinforcing the idea that the CNS is a HIV reservoir [66]. HIV escape in the CSF can be defined as HIV viral load in the CNS/CSF compartment exceeding detectable HIV RNA in the plasma [67,68,69,70,71]. CSF viral escape can be accompanied by symptoms such as progression of memory and executive dysfunction, cerebellar ataxia, or headaches [72]. Several factors are thought to lead to CSF viral escape including drug resistance, noncompliance with ART regimen, or poor CNS-penetration effectiveness (described below) [72]. Aside from the detection of HIV RNA and DNA in the CSF, HIV viral proteins (e.g., Vpr, Tat, gp120) have also been detected in the CSF [73,74,75,76]. Vpr, Tat, gp120, and Nef are demonstrated to be neurotoxic in animal models or in vitro systems [77,78,79,80,81,82,83,84] (also see the Adjunctive Therapies in Animal Models and In Vitro Systems section below).

### 2.5. Treatment

Since the advent of combined ART in the mid 1990′s, the incidence of HAD has dramatically decreased, although mild NCI is common in PWH, who are now living long enough to develop increasing numbers of comorbidities (see above) that likely potentiate the adverse effects of HIV infection in the brain [85]. Notably, multiple studies have shown that ART is associated with improved cognition in PWH who do not have virologic control [86,87,88] (Table 1). However, a major concern is that PWH who have mild NCI despite ART will be more prone to progress to dementia because of comorbidities and the effects of aging [85]. Therefore, treatment regimens that work to further reduce (or ideally eliminate) brain viral load and moderate the CNS immune response are being developed (see Future Treatment Directions below). ART treatment of PWH with NCI has been influenced by the categorization of ART agents by their CNS penetration-effectiveness (CPE). ART drugs are ranked 1 (lowest) through 4 (highest) in terms of molecular properties (protein binding or molecular weight), efficacy based on CSF HIV suppression, and known CSF concentration of the specific ART agent [89,90]. The efficacy of ART may depend on the target cell type where some ART agents have been shown to have higher half maximal effective concentration (EC_50_) in microglia in vitro, the putative HIV reservoir in the brain, vs. T cells or macrophages [91]. 

Asahchop and colleagues showed that the ART drug maraviroc, a CCR5 antagonist, is more effective at inhibiting HIV in mononuclear phagocytes (MPs) vs. T cells in vitro [91]. In a pilot study, the maraviroc treatment arm showed improved cognition in PWH [92] (Table 1). The CCR2 signaling pathway is also believed to be associated with cognitive dysfunction, as there is a correlation between CCR2+ monocytes and neurocognitive impairment in ART-untreated HIV-infected individuals [93]. An exploratory study on 17 virally suppressed HIV-infected individuals revealed that Cenicriviroc (CVC) (CCR2/CCR5 dual antagonist) also improved cognitive performance [94] (Table 1). Currently, it is unclear whether using ART with higher CPE scores can improve cognitive outcomes. Eighteen studies were reviewed regarding the relationship of CPE and cognitive performance. Eight studies concluded that a higher CPE score leads to better cognitive outcomes, three studies showed the opposite, while the remaining seven studies showed no relationship between CPE and improved cognition [95]. Few of these were controlled trials, but a small, randomized trial of 59 participants (some with virologic suppression at baseline, some without) did not show a cognitive benefit for CNS-targeted ART compared to non-CNS-targeted ART. More recently, a multi-national randomized controlled trial enrolled 191 people with HIV who had cognitive impairment and were taking suppressive ART [96] (Table 1). Participants were randomly assigned to intensify their existing ART regimen with a combination of (1) dolutegravir and maraviroc, (2) dolutegravir and placebo, or (3) dual placebo. Consistent with practice effect, cognitive performance improved in all three arms. However, the improvement in cognitive performance in the active treatment arms did not differ from that in the dual-placebo arm and thus did not support the conclusion that ART intensification benefits people with HIV who have cognitive impairment. Of note, the overall cognitive performance of the active treatment arms also did not decline, adding further evidence against the neurotoxicity of integrase inhibitors and maraviroc [96].

Unfortunately to date, many potential non-ART therapeutics tested in humans with HIV NCI have shown limited to no efficacy as well. In a 2016 clinical trial, lithium did not improve cognitive deficits or change MR spectroscopy results between treatment and placebo [97] (Table 1). Likewise, selegiline, another neuroprotectant with antioxidant properties, when given to PWH and NCI resulted in no significant cognitive improvement, and did not decrease oxidative stress biomarkers or result in beneficial changes in MRS parameters (e.g., a small rise in NAA/Cr in the BG and centrum semiovale or an increase Cho/Cr in the midfrontal cortex) [98,99] (Table 1). Sacktor and coworkers conducted a pilot randomized controlled trial to study monotherapy or combined therapy of a selective serotonin reuptake inhibitor, paroxetine, with an anti-fungal agent, fluconazole [100] (Table 1). This trial was based on results of paroxetine and fluconazole together having neuroprotective effects in vitro and protecting SIV-infected macaques from neuronal injury [101,102]. Paroxetine alone generally improved cognition in a battery of neuropsychological tests in PWH; however, its effects were not consistent over all cognitive domains tested and was not associated with consistent improvements in cellular stress markers and inflammation [100]. Fluconazole alone was associated with worsening in one of the neurocognitive tests and in CSF oxidative stress markers. The two treatments together were not associated with improvements in cognitive or laboratory tests compared to paroxetine alone. It is important to note that the authors go on to state in their Discussion that “Over the past 20 years, there have been over ten placebo-controlled trials of adjunctive agents to treat HIV-associated cognitive impairment, and none of them have shown improvement to enter clinical practice” [100].

Statins exhibit neuroprotective properties by reducing neuroinflammation in a rat model [103] and reducing monocyte chemotaxis in human peripheral blood monocyte culture [104]. Lipid metabolism dysregulation is associated with the pathogenesis of HAND [105]. The neuroprotective effect elicited by statins has potential for clinical treatment or prevention for HIV-1-associated neuropathogenicity. However, recent clinic studies indicated that statin use may not protect PWH from the development of NCI. Atorvastatin failed to significantly reduce neopterin (MP proinflammatory marker) and HIV RNA in CSF after high dose administration (80 mg) per day for 8 weeks (Table 1) [106]. Another clinical report with 1407 participants recruited in an AIDS cohort study, also did not indicate any benefit from statin treatment on cognitive performance [107].

Minocycline, an antibiotic with anti-inflammatory properties, was also hypothesized to have neuroprotective properties in HIV. However, a randomized controlled trial with minocycline failed to show a significant cognitive benefit in PWH with impairment despite effective ART [108,109]. Similarly, a randomized placebo-controlled trial testing the NMDAR antagonist memantine did not significantly ameliorate NCI in PWH [110]. However, memantine was associated with a significant increase in NAA/Cr (marker for neuronal integrity) ratios in the frontal white matter and parietal cortex, of mild NCI participants, by week 16 as measured by MR spectroscopy. Additionally, patients with lower baseline HIV RNA in the CSF exhibited even higher NAA/Cr in the frontal white matter compared to patients with higher baseline HIV RNA suggesting the importance of adjunctive use of ART [110].

In a randomized clinical trial, Marconi and coworkers showed that ruxolitinib (Rux), a JAK 1/2 inhibitor that is FDA approved for myelofibrosis and polycythema vera, is tolerated in PWH on ART [111] and decreased immune activation markers such as IL-18 and sCD14, which mechanistically relates to activated monocytes and have been implicated in trafficking to the CNS compartment [112,113,114]. Rux reduced CD4+/CD38+/HLA-DR+ cells that are associated with cell death and comorbid disease [111,115]. Rux can also reduce the number CD3+/CD4+/Bcl2+ cells, which are thought to maintain the latent viral reservoir [116]. Notably, Bcl-2 is a modulator of cell survival, and its upregulation may promote a further long-lived phenotype among cells harboring the viral reservoir. Downregulating this key cell survival marker, along with immune activation markers associated with disease progression, reservoir size, and systemic persistence including across the CNS, could provide a valuable therapeutic option to reverse immune dysfunction that prevents natural control of the latent reservoir, along with conferring a reduced lifespan of already-infected cells [117,118,119]. An initial concern was that combining an immunomodulator such as Rux may suppress the immune response resulting in viral recrudescence despite ART, although this was not found [111]. Recently, studies with a second generation JAK 1/2 inhibitor, baricitinib, which our team has also demonstrated confers anti-HIV effects including reversal of cognitive deficits in our murine model [120] (see Animal Models below), has been repurposed for COVID-19, garnering full FDA approval (first and only immunomodulator with this designation to date), a “strong recommendation from the World Health Organization; WHO), and first line treatment status worldwide. Studies with baricitinib have shown that baricitinib reverses immune dysfunction conferred by COVID-19, allowing the natural immune response to function without chronic inflammation from the virus, resulting in an increase in COVID-specific antibodies and concomitant restoration of immune activity [121,122,123,124]. These data underscore the delicate interplay between virally induced inflammation, functional immunity, and restoration of key immune functions in the presence of baricitinib in the viral infection setting. Future studies should explore whether immunomodulators such as baricitinib, which like Rux is a JAK 1/2 inhibitor but better tolerated, in combination with ART can improve performance in neuropsychological testing in patients with mild NCI (See Future Treatment Directions below).

### 2.6. Pathology

HIV encephalitis (HIVE) is typically defined as including HIV-infected cells within the brain, presence of multinucleated giant cells containing HIV antigens, which are overwhelmingly represented by MPs (i.e., macrophages and microglia), and also usually includes reactive or inflammatory glia (including MP and astrocytes) and neuronal abnormalities [125]. The neuronal abnormalities depend on the extent or severity of HIVE, but range from dendritic alterations to neuronal death [126,127,128]. Currently, virally suppressed patients with mild NCI do not exhibit apparent neuronal death, may or may not exhibit HIVE, and neuronal death and HIVE are not necessarily correlative factors for milder forms of NCI [125,129,130]. A study found that AIDS patients with progressive dementia exhibited no significant difference in neuronal loss when compared to HIV patients not diagnosed with dementia [131]. This suggests that more elusive mechanisms drive mild NCI in PWH. One possibility is that there is a smoldering infection accompanied by a persistent, low level of neuroinflammation, which drives synaptodendritic injury and scaling [132,133,134,135]. A neuronal synapse is composed of at least one axon (i.e., the longest process/branch of a neuron) projected onto a dendritic spine (i.e., bulbous protrusion on a shorter branch) of a neighboring neuron (Figure 1). The end of the axon is the presynaptic terminal while the dendritic spine is the postsynaptic terminal and both terminals are separated by a small space, the synaptic cleft. As an electrical signal travels down the axon (depolarization), this triggers the release of vesicles containing neurotransmitters (e.g., glutamate) at the presynaptic terminal, which can bind a cognate receptor (e.g., NMDA or AMPA) at the postsynaptic terminal. The strength of electrical conductance in a network of synapses, within the brain, is intrinsically linked to cognition, learning, memory, etc. which underlies everything discussed in this review [136].

In this review, synaptodendritic injury refers to dysfunction of synapses which may include reduction in receptors (e.g., NMDA or AMPA), decreased branching and length of dendrites, or disintegration of dendritic architecture such as loss of dendritic spine density or dysfunction of the neuronal cytoskeleton (Figure 1) [24,135,137]. Synaptic scaling is a specific type of reaction to synaptodendritic injury where synapses try to reestablish balance between inhibitory and excitatory synaptic transmission [135]. When neuronal calcium channels go awry and neurons are more susceptible to excitotoxicity, synapses usually scale down excitatory synapses (e.g., NMDA and upregulate inhibitory neurons e.g., GABA [135]. Synaptodendritic injury is often associated with HIV protein neurotoxicity (e.g., gp120) or from inflammatory cytokines and chemokines [24,138].

Notably, HIV proteins such as gp120, Tat, Vpr, and Nef can be detected in autopsy brains and have been associated with neurotoxicity [139,140,141,142,143]. In a recent report, Donoso and colleagues observed that residual production of HIV mRNA and proteins can still occur in latently infected cells (brain MPs and astrocytes) of brain autopsy samples despite long-term ART [144].

A more detailed review on the neurotoxicity of HIV proteins is beyond the scope of this review. An earlier autopsy study showed that synaptodendritic injury associated with NeuroHIV can correlate with the degree of cognitive impairment [145]. Similarly, Masliah et al. observed that dendritic injury in postmortem samples correlated with mild NCI [128]. The postmortem data suggests that disruption of synapses is probably linked to mild NCI in PWH.

Brain MPs are critical cell types driving NCI in PWH (Figure 1). In support of this hypothesis Gelman and colleagues examined autopsy samples from 140 infected human donors with mild NCI and the RNA and DNA pool present in brain tissue [63]. In this study, HIV RNA and DNA could be correlated with worse cognitive impairment if there was HIVE and microglial nodule encephalitis (MNE) in the brain (i.e., severe NCI). MNE is characterized as an encephalitis where activated microglial cells accumulate in nodules of varying sizes [125]. The authors concluded that reducing viral load through ART is more beneficial in improving cognitive function [63]. In contrast, in patients with mild NCI, without HIVE and MNE, the authors found minimal to no correlation between HIV RNA and DNA and neuropsychological test performance [63]. This suggests that viral replication alone is not the main driver of mild NCI in PWH.

In another postmortem study, tissue samples from PWH who had severe NCI (i.e., dementia), were usually found to have HIVE, which was defined as including HIV p24 immunopositivity and MP activation (Figure 1) [146]. The MP activity was intense as indicated by CD16, CD163, and MHCII staining in the brain parenchyma and perivascular regions [146]. Furthermore, HIVE included macrophage/microglial nodules showing morphology consistent with pronounced activation. In mild NCI postmortem brain autopsy samples typically there was an absence of HIV immunoreactivity, and MPs were less activated, although they were still more activated than control brain autopsy specimens [146]. This suggests that despite HIV suppression with ART, MP activity is an important component in mild NCI in PWH.

Ginsberg et al. further extended the findings by Tavazzi and colleagues by performing a microarray analysis on autopsy samples from HIVE (mostly severe NCI), PWH without HIVE (mostly mild NCI), and HIV-negative donors [147]. Briefly, the study isolated MPs from the brain parenchyma of the three groups and analyzed >500 transcripts related to inflammation, cellular stress, and apoptosis. Proinflammatory cytokines such as IL-6 and the primary TNFa receptor (*Tnfrsf1a*) were upregulated in HIVE and HIV+ without HIVE samples. Importantly, while the study outlines that the HIV+ mild NCI group upregulates macrophage/microglial transcripts to a lesser degree compared to the HIVE (i.e., severe NCI) group, it also further highlights the role that brain MPs play in HIV infection in the CNS as it relates to mild NCI [147].

## 3. Adjunctive Therapies in Animal Models and In Vitro Systems

In this section we will briefly review animal models that resemble HIV NCI and/or HIVE found in humans. There are no perfect animal models for any disease, including those for the study of HIV NCI/HIVE. Nevertheless, animal models are typically used and, for the most part, required for preclinical testing of novel treatments. While it is not possible to review all HAND animal models and their associated literature, we have selected publications that are relevant to mild HAND in humans and/or preclinical treatment trials that are relevant to the main themes of this review. In another subsection below, we will briefly review compounds that were tested in vitro; however, some of these studies tested potential adjunctive therapies in an animal model and in vitro simultaneously. Therefore, the two sections were combined.

### 3.1. Animal Models

One model for studying NCI associated with HIV is a HAND model that utilizes severe combined immunodeficiency (SCID) mice [148]. In this model, human monocyte derived macrophages (MDMs) are infected with HIV and then intracranially (IC) injected into the right frontal cortex and BG of SCID mice [148]. Although a short-term model, the model is suitable for screening novel HIV therapeutics in the brain. Importantly, mice injected IC with HIV-infected MDMs (and not controls injected with uninfected MDM) exhibit mild NCI during behavioral testing recapitulating what is observed in human patients on ART [120,137,149]. Also significantly, studies using this model have shown that typical combination ART does not eliminate HIV in the brains of these mice, although admittedly treatment was short term [150,151,152].

Based on clinical studies showing that CSF interferon-alpha (IFN-α) levels correlate with NCI in PWH [47,50], Koneru and colleagues tested a biologic, B18R, which strongly binds IFN-α (Table 2) [137]. Groups of mice included control/saline, HIV + ART, and HIV + ART + B18R, and HIV + saline. The combined ART and B18R treated HIV mice showed astrogliosis and microgliosis levels similar to control mice [137]. Additionally, dendritic branching in the HIV/ART/B18R group was similar to the control group, unlike the HIV mice treated with ART alone or HIV/Saline groups [137]. Finally, cognitive function in HIV mice treated with ART plus B18R showed comparable performance to control mice [137]. The data from this study suggests the importance of reducing viral load (via ART) and mitigating inflammation (via B18R) simultaneously when treating NCI associated with HIV.

The natural polyphenol curcumin, isolated from *Curcumin longa* (turmeric), has been documented as an anti-inflammatory and antioxidant agent in the context of diseases like multiple sclerosis, Parkinson’s disease, and Alzheimer’s disease (Table 2) [153]. Using the SCID mouse model described above, mice intracerebrally injected with HIV-infected MDMs and subcutaneously treated with soluble curcumin (on day 6 after behavioral testing) showed a reversal of cognitive impairment during the object recognition test (ORT) on day 13 (Figure 2) [154]. In the same preliminary study, there was a downward trend of cells double positive for the myeloid activation markers MHCII and CD11b (i.e., activated MP), as measured by flow cytometry, in curcumin treated mice vs. HIV infected mice. Improvements in both cognition and MP activation are important components in the pathogenesis of HIV associated NCI and suggest that curcumin could be considered for investigation in PWH and NCI.

Other promising therapeutics studied in the above described SCID HAND model include the Jak 1/2 inhibitors ruxolitinib and baricitinib that function as “immunomodulators” [120,155] (Table 2). Ruxolitinib was found to decrease activation markers of the IC injected human MDMs in vitro [155]. Furthermore, ruxolitinib reduced astrogliosis, mouse microglial activation, and p24+ cells in HIV-infected brains, suggesting that ruxolitinib has both antiviral and anti-inflammatory activity [155]. In a more recent study, baricitinib reversed NCI in the HIV-infected mouse group [120]. Like ruxolitinib, baricitinib diminished activation of human MDMs in vitro. Additionally, brain pathology of HAND mice revealed that baricitinib reduced mouse astrogliosis, mouse microgliosis (CD45/MHCII) and reduced brain viral load, also suggesting it has antiviral and anti-inflammatory effects [120] (Table 2). These studies laid the foundation for investigation of Jak 1/2 inhibitors in conjunction with ART in human patients (see Future Treatment Directions below).

The EcoHIV murine model is generated by utilizing HIV that is pseudotyped with the envelope glycoprotein gp80 from ecotropic murine leukemia virus [156]. This novel chimeric virus establishes persistent infection in mouse peritoneal macrophages, brain myeloid cells, CD4+ T cells, and splenic lymphocytes [156,157,158]. In a 2019 study using this model, intranasal insulin was found to ameliorate cognitive impairment and dendritic injury [159] (Table 2). EcoHIV infected mice also demonstrated a downregulation of important gene transcripts such as calmodulin dependent protein kinase II (CaMKII), neurogranin, and brain-derived neurotrophic factor (BDNF) necessary for neuronal health and maintenance. CaMKII and neurogranin are part of the calcium signal transduction pathway important for postsynaptic function and maintenance whereas BDNF is a critical factor for neuronal development, differentiation, and maintenance. The EcoHIV group treated with insulin showed similar mRNA expression of neuronal genes assayed when compared to controls [159]. Intranasal insulin also appears to demonstrate an antiviral effect as HIV was decreased in the treatment. Unfortunately, the authors documented that after treatment cessation, NCI is similar to the untreated EcoHIV group after 24 h, suggesting that insulin’s effects are short-lived.

A recent study using the EcoHIV model tested a glutamine antagonist JHU083 [160]. Nedelcovych and colleagues found that JHU083 can normalize glutamate levels in the CSF and in microglia in their model [160]. Excess glutamate is thought to be associated with excitotoxicity and neuronal dysfunction, linked to important NMDA and AMPA receptors involved in cognition, during HIV infection in the CNS [160]. As excitatory neurons compose most neurons in the brain, JHU083 presents a potential treatment option for clinical trials.

In another recent study using the EcoHIV model, a novel peroxisome proliferator-activated receptor-γ (PPARγ) agonist was investigated (Table 2) [161]. PPARγ are a category of nuclear receptors that play a pivotal role in lipid/cholesterol metabolism and glucose homeostasis [162]. PPAR-gamma isoform can be found on macrophages and agonists to the receptors show anti-inflammatory and antioxidant effects in models for multiple sclerosis and Parkinson’s disease [163,164]. The PPAR-gamma agonists such as rosiglitazone are thought to modulate the activation of nuclear factor kappa B (NF-κB), a key transcription factor for cytokine production [165]. While PPARγ agonists have been studied in HIV NCI, and are clinically available for the treatment of diabetes, side effects such as cardiovascular toxicity, edema, and weight gain may discourage their use. Omeragic et al. observed that the newly discovered PPARγ agonist INT131 could downregulate inflammatory markers such as TNFα, IL-1β, CCL3, and C3 in the EcoHIV infected group [161]. The side effects described above were not observed in these mice and INT131 was shown to cross the blood brain barrier. While this group did not investigate whether INT131 had antiviral properties, their previous in vivo study using PPARγ agonists in the same drug class have demonstrated anti-HIV and anti-inflammatory properties by decreasing mRNA levels of TNFα, IL-1β, CCL2, CCL3, CXCL10 and HIV p24 protein in the murine brain [166]. If future studies show that INT131 has anti-HIV properties, combined with its anti-inflammatory effects, these important effects highlight another potential future treatment option that may be combined with current ART, where ART alone is insufficient to prevent or reverse HIV NCI. In an earlier study by Omeragic et al., the PPARγ agonist rosiglitazone reduced TNF-α and IL-1β mRNA and excitotoxicity in a gp120 neurotoxicity mouse model, thus supporting further investigation with PPARγ agonists [167].

Another promising murine model is the bone marrow–liver–thymus (BLT) humanized mouse model, which include the human myeloid predominant (MOM) and human T-cell predominant (ToM) models [91,168,169,170]. In this model, NOD/SCID mice are injected with human bone marrow, liver, and thymus. The mice are also reconstituted intravenously with human CD34+ hematopoietic stem cells allowing for the circulation of human myeloid cells and/or lymphocytes. Honeycutt et al. found that human macrophages are present in brains using the MoM BLT model [168]. The authors also showed p24+ staining in the brain following intravenous HIV infection [168]. In a parallel study, Asahchop and colleagues also found human T lymphocytes and macrophages in brains of both the HIV infected untreated BLT mice group and HIV infected BLT mice treated with ART [91]. Importantly, HIV p24 positive cells were undetectable in the brain following ART treatment. Likewise, HIV DNA and RNA was undetectable in the brains of HIV BLT treated mice using digital droplet PCR following ART [91]. However, upon ART interruption in BLT mice: HIV DNA, HIV RNA, and Human CD68 RNA is detectable in brain tissue. While the BLT model is an attractive model in the field, it is unclear whether HIV infected BLT mice develop cognitive dysfunction, the most important clinical manifestation of HIV brain infection. Preliminary data in our lab suggest that 18-week-old (NOD/SCID) BLT mice infected intraperitoneally with HIV ADA develop cognitive dysfunction when tested with serial object recognition tests (ORT) over time and finally using water radial arm maze (WRAM) testing after a 3-month infection period (Zhang et al. unpublished).

Recent studies suggest that the shock-and-kill approaches may lead to excessive neuroinflammation, which could result in bystander CNS damage associated with HAND [171,172]. Instead, a strategy called “block-and-lock” provides an alternative therapeutic approach in suppressing viral expression in the CNS. The viral protein Tat (trans-activator of transcription) recognizes the transactivation response element (TAR), and activates the assembly of the transcription complex, and initiates transcription elongation. Didehydro-Cortistatin A (dCA) serves as a Tat inhibitor, which inhibits viral transcriptional activity and promotes latency establishment. HIV-infected humanized mice treated with dCA showed significantly reduced HIV RNA in the CNS [173] (Table 2). Transgenic mice that constitutively express Tat in astrocytes develop astrocytosis [80], as Tat triggers aberrant inflammatory cytokine expression including IL-1β and TNF-α in the CNS. A recent study demonstrated that dCA reverses the Tat-mediated cytokine dysregulation in vitro, which suggested that dCA should be considered as a promising therapeutic agent for the treatment of NCI in PWH [174].

The accessibility of rodent models that can be genetically manipulated led to the generation of transgenic models that express HIV proteins in brain tissue. For example, HIV Tat or gp120 can be produced in astrocytes, or Vpr can be expressed from myeloid cells in the CNS [77,80,82]. The advantage of transgenic models allows investigators to isolate the neuropathological effects of a specific HIV protein; however, it can be difficult to interpret a sole protein’s effect in relation to the pathogenesis of human brain HIV infection where other HIV proteins and numerous inflammatory factors are present and lead to the summation of neurotoxic effects during HIV brain infection. Regardless, transgenic models are useful in studying adjunctive therapies (see paragraph above or described below). Other genetic manipulations of rodent models may include knocking out a particular gene (e.g., PD-L1; described below) allowing researchers to investigate mechanisms behind HIV reservoir establishment and potential treatments.

In a gp120 transgenic mouse model (gp120-tg), an immunomodulatory drug (FK506) was shown to reverse dysfunction from HIV infection [175] (Table 2). FK506 was shown to decrease the inflammatory markers GFAP, Iba-1, and IL-6. Furthermore, FK506 reverses the neurotoxicity of gp120 by restoring the neuronal markers MAP2 and synaptophysin, indicators of properly functioning synapses. The immunomodulatory and neuroprotectant functions of FK506 are traced to its modulatory effects on mitochondrial function [175].

HIV-1 DNA, RNA, and protein in microglial cells can be detected from post-mortem brain tissues [176]. Microglia are intrinsically resistant to CTL-mediated killing [177]. This resistance to killing can be attributed to high programmed cell death PD-L1 expression in microglia [178]. The PD-1/PD-L1 pathway serves as an immune checkpoint modulator, which delivers inhibitory signaling that can negatively regulate CTL immune responses [179]. In order to investigate whether PD-1/PD-L1 is responsible for inhibiting cytolytic responses, a heterologous prime-CNS boost mouse model was used to investigate CD8^+^ T cell-elicited cytolytic responses against HIV-1 peptide-presenting microglia from wild-type C57BL/6 and PD-L1 KO mice [180]. The study tried to determine PD-1/PD-L1-elicited inhibition against cytolytic responses. Mice were primed with an adenovirus vector constitutively expressing the HIV-1 p24 capsid protein followed by a CNS boost with HIV virus-like particles (HIV-VLPs, carrying Gag and Env proteins) via intracranial injection into the striatum [180]. This procedure allows the establishment of resident CD8^+^ T cells that are specific for HIV-1 Gag epitopes in the brain [180]. Later, microglia pulsed with Gag epitopes were injected intracranially, and the susceptibility of microglia to cytotoxic T-lymphocytes killing activity was assessed [180].

Interestingly, PD-L1 knockout mice displayed resistance to CTL-mediated killing of microglia compared to wild-type mice [180]. Blocking the PD-1/PD-L1 pathway may be beneficial in reducing the brain viral reservoir in microglia. A phase 1 clinical trial is currently underway to determine whether pembrolizumab (PD-1 inhibitor) is safe and tolerable in PWH (ClinicalTrials.gov ID: NCT03239899) [181].

Aside from rodent models, HIV infection of the CNS has been modeled using feline immunodeficiency virus (FIV) [182,183]. Replication-competent FIV is recoverable from feline brain tissue and, like HIV in human brain, FIV prefers to infect feline microglia [184,185]. Importantly, this model allows one to study long term retroviral brain persistence and, in addition and similar to the human condition, FIV infection affects motor and cognitive function when correlated with viral burden and inflammation in the feline brain [186]. Relevant to all important NeuroHIV animal models, neurobehavioral testing demonstrated FIV infected felines were more impaired than uninfected cats [186]. Additionally, Maingat et al. observed that the AMPA receptor (AMPAR) and Kainate receptor were reduced in FIV+ brain tissue [186]. The authors linked this observation to decreased D-serine levels, a co-agonist for the glycine binding site on NMDA receptors (NMDAR), leading to a glutamate/glutamine imbalance in the feline brain [186].

Interestingly, ART drugs such as tipranavir, zidovudine, lamivudine, and didanosine have been shown to improve cognitive and motor function in the FIV model [187,188,189]. In 2013, intranasal insulin was shown to be effective in Phase II clinical trials for Alzheimer’s disease [190]. Mamik et al. later observed that intranasal insulin had both anti-inflammatory and antiviral properties both in vitro and in their FIV model, which was also tested in the EcoHIV model published by Kim et al. in 2019 [159,191] (Table 2). Mamik and colleagues saw a reduction in IL-6 in the cortex and a reduction in CXCL10 in both the cortex and striatum regions of the feline brain. They also showed a reduction in FIV RNA in the cortex. Neuropathological analysis of intranasal insulin treated animals showed intact neuronal processes vs. untreated FIV-infected animals [191]. Nissl staining also confirmed that intranasal insulin impeded neuronal loss in the cortex, striatum, and hippocampus. Lastly, intranasal insulin improved neurobehavioral performance in FIV-infected animals [191]. Insulin treatment can serve as a potential treatment option for NCI linked to NeuroHIV infection due to its suggested anti-inflammatory and antiviral characteristics. Insulin in combination with ART is an interesting potential treatment option considering it has been shown that intranasal administration of insulin is beneficial to Alzheimer’s disease patients in phase II trials [190].

The use of simian immunodeficiency virus (SIV) infected rhesus macaques has been used extensively to model HIV infection systemically. SIV models have also been used to study SIV infection in the CNS. CNS SIV models include the pigtailed macaque (*Macaca nemestrina*) model, and the CD8+ depletion *Rhesus macaque* model. In the pigtail macaque model, the animals are infected with the SIV/17E-Fr molecular clone followed by coinfection with CD4+ T-cell depleting SIV/DeltaB670 strain [192,193]. In the CD8+ depletion model, animals are infected with SIVmac251 followed by transient depletion of CD8+ cells by an anti-CD8 monoclonal antibody given 6, 8, and 12 days after infection [194]. The authors saw that within 3–4 months, more than 85% of infected macaques develop SIV encephalitis (SIVE), if consistently depleted for CD8+ for more than 28 days [194].

In both SIV models, ART regimens have been successfully tested and have been shown to halt SIVE progression, yet these SIV-infected monkeys still have persistent neuroinflammation [54,195,196,197,198]. SIV macaque models have been used to study biomarkers in the CSF and plasma that are associated with SIV infection in the CNS [199,200,201]. The pigtail macaque model has been used to study systemic and CNS SIV persistence through a viral outgrowth assay, making it a suitable model to examine HIV cure strategies [202]. One adjunctive treatment studied in the SIV model was minocycline [203,204,205] (Table 2). In these studies, minocycline was demonstrated to reduce SIV viral load in the CNS and decrease the recruitment of monocytes/macrophages and their activation (e.g., MHCII, CD11b, CD163) [203,205]. Also following minocycline treatment, immunohistochemistry analysis for markers of synaptodendritic injury and glial activation (microgliosis and astrogliosis) were prevented compared to untreated SIV-infected animals [204]. Furthermore, MRS of minocycline treated animals showed preservation of neuronal integrity measured by NAA/Cr ratio [204]. While these studies did not examine minocycline treatment in conjunction with ART, minocycline showed promise as a treatment for HIVE due to its antiretroviral, anti-inflammatory, and neuroprotective characteristics [203,204,205]. Unfortunately, in a randomized placebo-controlled clinical trial in PWH, minocycline did not improve cognition or influence any measured CSF markers, except for a curtailment in CSF ceramides, biomarkers for oxidative stress [108,109] (Table 1).

HIV-infected monocytes can migrate across the blood–brain barrier (BBB) where they establish or reseed the CNS [206]. Rhesus macaques (*Macaca mulatta*) were employed to evaluate the impact of MNP trafficking on SIV-mediated neuropathogenesis [207]. Administration of the anti-α4 integrin antibody (natalizumab) successfully blocked the trafficking and accumulation of SIV infected MNPs in the brain (Table 2); meanwhile, the blockage of viral reseeding also stabilized neuronal injury from ongoing SIV infection [207]. Therefore, the blockade of MP activation and migration towards to the CNS can provide a potential therapeutic intervention against reservoir establishment, reseeding and neuronal injury.

Unfortunately, cognitive testing of non-human primates is sparse in the SIV literature [54]. One study has shown that untreated *Rhesus macaques* exhibit severe NCI comparable to what it seen in untreated humans [208]. Weed et al. adapted the Cambridge Neuropsychological Test Automated Battery (CANTAB) for Rhesus macaques which is used in humans to infer cognitive impairment associated with the temporal cortical, frontal cortical, and subcortical brain regions [208]. The authors observed that untreated SIV-infected monkeys performed poorly in tasks that tested spatial working memory, reaction time, and fine motor skills linked to the frontal cortical and subcortical regions [208,209,210,211]. In general, SIV-infected macaques did not perform poorly in a task to measure recognition memory associated with the temporal cortical region [212,213]. The results of the study probably recapitulate cognitive dysfunction seen in AIDS patients where the frontal cortical and subcortical is impaired [214,215,216,217]. However, neurobehavioral testing used for humans is difficult to experimentally implement with non-human primates, a factor that at least partially explains the dearth of cognitive studies using SIVE models [54,218].

### 3.2. In Vitro Studies

The use of in vitro systems allows investigators to study the mechanisms of novel adjunctive therapies for NeuroHIV in a more controlled manner. Researchers may take a bottom-up approach and investigate new agents in tissue culture systems perturbed by HIV infection to examine possible novel therapeutics effects. Investigators may also take a top-down approach and explore drugs that are approved for human use in other brain diseases and test them in vitro or in animal models for HIV NCI. Repurposing existing FDA approved compounds for other brain diseases that share overlapping pathways known to be disrupted by HIV infection can be advantageous when considering future clinical trials for HIV NCI. Uncovering specific mechanisms related to how a novel therapeutic agent protects against HIV infection or HIV protein neurotoxicity in vitro may be informative for future animal studies and human clinical trials. While in vitro studies should preferentially be performed with primary human cells, we acknowledge that it is not feasible in some cases. We specified the in vitro source for each selected study in Table 2 below.

The FDA-approved drug dimethyl fumarate (DMF) has been shown to be an immunomodulatory compound in clinical trials for multiple sclerosis [219,220]. Several in vitro studies, have shown that DMF can inhibit HIV replication in human monocyte-derived macrophages (MDMs), reduce the accumulation of macrophages via CCL2 signaling, attenuate inflammation, and dampen the production of neurotoxins of human MDMs when conditioned media was incubated with primary rat cortex neurons [221,222] (Table 2). With perivascular macrophages and microglia being the relevant cells infected by HIV, and prime producer of neuroinflammatory and neurotoxic molecules, DMF was investigated in a macaque model [223]. While the authors observed that DMF can induce an antioxidant response in macaque brains, they did not observe an inhibition of SIV replication, signs of reduced inflammation, or a change in synaptic protein expression in DMF-treated non-human primates. It should be noted that CD8+ depletion macaque model in this study presented the hallmarks of severe neuropathology as seen with PWH with severe NCI [223]. It is still an open question as to whether DMF could function as an adjunctive therapy with ART in a SIV macaque model with milder NCI.

Host transient expression of interferon-beta (IFN-β) has been documented to delay the progression of SIV infection in macaques [224,225]. The anti-inflammatory action of exogenous IFN-β was tested in vitro and in a gp120 transgenic mouse model. The authors found that IFN-β could protect primary rat and murine neurons from gp120-induced neurotoxicity in a dose dependent manner [83]. When neurons were incubated with a neutralizing antibody to CCL2 (monocyte chemoattractant), IFN-β treatment was ineffective. Thaney et al. then knocked out the Type I interferon receptor and observed that exogenous IFN-β did not protect neurons from gp120 neurotoxicity. The in vitro data suggests that CCL2 and the Type I interferon receptor are mediators of IFN-β neuroprotection against gp120 neurotoxicity [83]. Extending their in vitro findings, the authors also observed that intranasal administration of IFN-β could reduce the activation of microglial cells in the hippocampus and cortex of a mouse model that expresses gp120 in the brain. Additionally, synaptodendritic damage was reduced [83]. Systemically administered IFN-β can reduce brain inflammation and is an FDA approved treatment for multiple sclerosis [226,227].

HIV proteins such as Tat and Vpr can induce NLRP3 inflammasome activation [74,228]. A recent in vitro study has also shown that even single-stranded RNA from HIV’s long-terminal repeat region can activate the NLRP3 inflammasome in primary human microglia [229]. Inflammasome activation leads to IL-1β and IL-18 release linked to chronic neuroinflammation [230]. Furthermore, transcripts of the inflammasome associated cytokines IL-1β and IL-18 are upregulated in in the cerebral white matter of HIV+ postmortem brains relative to uninfected donors [231]. The NLRP3 inflammasome presents itself as another therapeutic target; therefore, He and colleagues tested a novel small molecule inhibitor (MCC950) of the canonical NLRP3 inflammasome both in vitro and in a gp120 transgenic model [232] (Table 2). The authors found that treatment of BV2 cells (microglia cell line) and primary mouse microglial cells with MCC950 curtailed the production of tumor necrosis factor alpha and the neurotoxin nitric oxide. Furthermore, the study found that MCC950 could protect neurons in vitro from dendritic injury when exposed to conditioned media from HIV gp120 activated microglia [232]. By extension, MCC950 treatment of mice that express HIV gp120 in the CNS exhibited less activated microglia, reduced synaptodendritic injury in the cortex and hippocampus, and improved performance in behavioral testing [232]. Again, another intriguing compound that can be further examined as an adjunctive therapy given its therapeutic properties briefly outlined above.

During some types of neuronal injury, a buildup of misfolded protein in the endoplasmic reticulum (ER) can lead to ER stress and the unfolded protein response. The unfolded protein response is mediated by trans-ER membrane proteins such as the protein kinase RNA-like ER kinase (PERK), which represents a critical therapeutic target that is associated with ER stress, neuroinflammation, and neurodegeneration [233,234,235,236]. When a mouse microglial cell line (BV-2) was exposed to HIV Tat, the PERK inhibitor GSK2606414 was able to decrease levels of inflammatory mediators such as TNF-α, IL-6, induced nitric oxide synthase, nitric oxide, and monocyte chemoattractant protein 1 [237] (Table 2). Future studies of PERK inhibitors, in neuron cultures involving broader HIV related neurotoxicity mechanisms, and in HIV animal models that measure potential effects on NCI, could prompt the investigation of PERK inhibitors in PWH.

In a 2015 study, Steiner et al. screened over 2000 drugs that were protective against oxidative stress that led to neuronal injury. The authors narrowed the screen down to several selective serotonin reuptake inhibitors, namely paroxetine [101]. In primary rat neuron cultures exposed to HIV neurotoxic proteins (e.g., Tat and gp120) and toxins that induced oxidative stress or excitotoxicity, paroxetine was able to protect mitochondria from calcium-induced excitotoxicity, neuronal death, and modulate inflammatory cytokine production such as IL-1α, IL-1β TNF-α, or IL-6. Additionally, paroxetine was found to be neuroprotective in a gp120 neurotoxicity rat model and could stimulate growth of neuroprogenitor cells in vitro and in their animal model [101]. Paroxetine was ultimately tested in a randomized placebo-controlled clinical trial, as outlined in the Treatments section above, but its success was limited in ameliorating NCI in PWH [100].

Lithium has been used to treat psychiatric disorders such as bipolar disorder for decades [238] and explored in pilot clinical trials for neurodegenerative diseases such as Alzheimer’s disease [239,240,241]. Two earlier HIV studies examined the therapeutic benefit of lithium and found it to be protective in a rat model and human neuronal cell line (SH-SY5Y) against gp120 toxicity and in primary rat neurons exposed to Tat toxicity [242,243]. However, lithium was administered in a randomized placebo-controlled trial for PWH and NCI mentioned in the Treatments section above, but the authors did not observe improved cognitive outcomes in the lithium treatment arm compared to placebo [97] (Table 1).

The accumulation of amyloid plaques and Tau containing neurofibrillary tangles in the brain, which is concomitant with progressive neurodegeneration, suggests that maintenance of normal neuronal autophagy is disrupted. HIV-induced neurodegeneration is also associated with autophagy dysfunction in neurons, where viral proteins, such as Tat and Nef, can inhibit autophagy and induce autophagosome degradation [244,245]. Rapamycin, which serves as an autophagy inducer, successfully ameliorated the HIV-induced neurotoxicity in cultured primary fetal neurons [246] (Table 2). Notably, the therapeutic strategy to enhance the autophagy pathway in neurons may be a double-edged sword in that autophagy plays a critical role in various biological functions, including cell apoptosis, immune surveillance, cytokine production, stress response, mitochondrial metabolism, etc. [247]. However, enhancing neuronal autophagy without interfering in homeostasis of bystander cells would be difficult.

A recent study focusing on ART with long-term viral suppression demonstrated that CD14+ CD16+ monocytes carrying integrated HIV, which circulate in peripheral blood, can migrate across the BBB in a CCL2 dependent manner [248]. Therefore, infiltrating MPs and microglia are considered major CNS viral reservoirs and sources of viral persistence. The increased junctional protein expression including junctional adhesion molecule A (JAM-A) and activated leukocyte cellular adhesion molecule (ALCAM) facilitate the chemoattractant CCL2-mediated transmigration of the MNPs across the BBB [206]. León-Rivera and colleagues discovered that anti-JAM-A, anti-ALCAM, or CCR2 inhibitor application successfully blocked/reduced HIV-infected monocytes transmigration in a human BBB model [248] (Table 2). Therefore, CCR2 and its related junctional proteins, JAM-A and ALCAM, have been proposed as potential therapeutic targets to prevent CNS viral reservoir reseeding.

The ineffective penetrance of antiretroviral drugs across the BBB is a factor contributing to CNS persistence of HIV and heightens the risk of HIV-related NCI. Contemporary ART regimens (zidovudine (AZT), etravirine (ETR), raltegravir (RAL), darunavir (DRV), maraviroc (MVC) and dolutegravir (DTG)) have low efficacy and concentration in the brains of BLT mice compared to ART effects in peripheral blood [91]. HIV-1 RNA is still detectable in the CSF after years of suppressive ART in human cohorts [249]. Novel nano-formulated ART has the potential to improve the therapeutic efficacy in infected MPs within the CNS. The excellent biocompatibility of nanodiamond (ND) formulations makes them a potential ideal candidate for drug delivery to the CNS. Nanodiamond formulation conjugated with efavirenz (EFV) showed improved pharmacokinetics with respect to prolonged retention, enhanced delivery capacity and sustaining viral suppression in HIV-infected macrophages in vitro [250] (Table 2). There is evidence that efavirenz is neurotoxic, therefore, other ART compounds should be considered for ND formulations and other future nanocarrier studies [251]. Another group using poly (D, L-lactide-co-glycolide) (PLGA) encapsulated elvitegravir (EVG), successfully improved drug penetration and viral suppression within MDMs [252] (Table 2). Using a xenografted mouse model, whereby HIV-infected MDMs are injected intracerebrally into basal ganglia, PLGA-based nanocarrier facilitated the accumulation of EVG in brain tissue, which led to significantly reduced viral burden in the CNS [253].

Nucleoside reverse transcriptase inhibitors (NRTIs) can interfere with ATP production and interrupt DNA replication in neuronal mitochondria, which can result in neurotoxicity [254]. And like most ART, NRTIs have poor BBB permeability [255]. Nanogel, used in a nano-NRTI formulation, reduced neurotoxicity by lowering reactive oxygen species and apoptosis in cultured rat neurons and simultaneously reducing viral expression in mouse brain [256] (Table 2).

## 4. Future Treatment Directions

Human clinical studies indicate that NCI in PWH remains common even in those receiving highly active ART with relatively effective CPE scores (i.e., better BBB penetration) [89,90]. In addition, a significant proportion of these patients can show progression of cognitive dysfunction despite ART [85] (see Section 2.1 Cognitive subsection of the Clinical and Translation section above). That is because HIV persists in the CNS and this is supported by the preponderance of CSF and autopsy studies of PWH receiving ART [65,67,68,69,70,71]. There is concern that persistent brain HIV combined with an aging population of PWH on ART will result in a larger proportion of dementia, particularly as HIV may interact with common comorbidities that impact NCI, such as depression, early Alzheimer’s disease, stroke, traumatic brain injury, and others [85]. In addition, most animal model studies employing ART suggest that retroviruses cannot be eliminated from the brain using current conventional approaches [150,151,152], although more recent data using humanized mice, which harbor brain HIV for months and given prolonged ART, suggests elimination may be possible [91].

The lack of success in multiple human trials using mild NCI as a primary endpoint of treatment efficacy could be due to factors aside from absence of drug efficacy. Recruiting PWH with NCI who also do not have comorbidities that potentially contribute to NCI (and thus ‘muddy the water’) can be challenging. In addition, it is well known that mild NCI in PWH can remain stable, improve or progress [257,258]. If the novel drug being studied reverses mild NCI, then it may be possible to recruit enough patients to demonstrate a statistically significant effect, provided the effect can be seen within the treatment time being studied. An emerging potential surrogate is cell associated RNA and DNA in the CSF. A previous study has demonstrated that CSF HIV DNA in the setting of virologic suppression is detectable in approximately 50% of the study’s participants, a finding that has been associated with NCI impairment [259]. Even more recent research with novel assays that detect the LTR has shown cell associated HIV RNA and DNA is detectable in 80–90% of people on ART [260]. However, it is not clear if changes in such surrogates will be associated with clinical improvement. Other surrogate markers, such as CSF immune parameters and imaging techniques can also be used [24,51]. Clinical studies may be most informative if both cognitive and biomarker surrogates including viral, immunologic, and imaging are included.

Many animal models and in vitro studies have shown that certain HIV proteins, including Tat, gp120, Vpr and Nef, are neurotoxic (i.e., produce adverse effects on neuronal structure and function) [77,78,79,80,81,82,83,84]. If more effective ARV (antiretroviral) strategies can be shown to reduce or even eliminate CNS HIV load, then consequently individual HIV proteins that have neurotoxicity will also be reduced. However, given that current ART does not eliminate brain HIV, the practical approach to reducing individual HIV proteins in human brain is more difficult to address. Delivering multiple, individual neutralizing antibodies to each neurotoxic HIV protein is impractical. One potential approach discussed above is using dCA to specifically inhibit Tat, which not only might reduce Tat induced neurotoxic effects, but reduce HIV load [173] (Table 2). Somewhat similar studies to the dCA treatment of Tat, which examined treatments for specific HIV neurotoxic proteins were reviewed above and could be moved forward to human clinical trials if tested in an appropriate animal model of retroviral infection that demonstrates improvement or prevention of NCI and/or reduction in brain HIV.

Thaney et al. studied treatment of rodent neurons exposed to gp120 and showed neuroprotection using interferon-beta (IFN-b) [83] (Table 2). They then used intranasal IFN-b to treat gp120 transgenic mice and showed that treatment reduced the activation of microglia and reduced synaptodendritic damage in the hippocampus. Systemically administered IFN-b has been used in patients with multiple sclerosis for 30 years and is reasonably well tolerated [226,227]. If this treatment can be shown to be effective in a model of whole retrovirus brain infection, it could be considered for a treatment trial in PWH.

Similarly, the inflammasome inhibitor, MCC950, was tested using in vitro models of gp120 neurotoxicity and the gp120 transgenic model [232] (Table 2). Microglial activation and synaptodendritic injury were reduced, again suggesting that if this compound is tested in an animal model of retroviral infection and has similar effects, it could be moved forward to human clinical trials.

Other novel approaches to reduce brain HIV load include those using improved CNS delivery systems (Table 2). Many ARVs have poor BBB penetration. Using nanodiamond or nanogel formulations, as discussed above, may provide improved CNS delivery and further accordingly further reduction or elimination of HIV brain load compared to conventional ART [250,256]. However, these delivery systems need to be tested in appropriate animal models for demonstration of CNS HIV reduction or elimination prior to human trials. Also as discussed above, PLGA-based nanocarrier has been used in a xenograft mouse model and reduces CNS virus [253]. Assuming this could be demonstrated to be safe in humans and without CNS ARV toxicity, it would represent a valuable advancement in HIV associated NCI treatment.

Blocking leukocytes, including both HIV-infected CD4+ T cells and monocytes, from crossing the BBB could prevent HIV reseeding of the brain and possibly reduce the incidence of HIV-induced brain disease [248]. However, a major concern would also be potentially blocking important cells from entering the brain that are responsible for ‘immune surveillance’. For example, natalizumab treatment (Table 2), which binds alpha 4 integrins and is used to treat people with multiple sclerosis, is associated with a significant rate of progressive multifocal leukoencephalitis (PML), a disease caused by JC virus, which can be fatal [261]. One of the theories of PML evolving in these patients is that the lack of immune surveillance due to exclusion of lymphocytes by natalizumab treatment results in the lack of suppression of JC virus in the brain. Thus, preventing important immune cells from entering the brain during HIV infection could exacerbate CNS HIV and worsen brain disease.

Various novel treatments that have been pursued in preclinical studies have addressed some of the proposed pathogenic mechanisms of HIV induced brain disease in PWH. As partially reviewed above under Treatment Section 2.5, some of these studies have prompted human treatment trials as add on therapies to ART [97,98,100,105,106,111]. Unfortunately, none of these small studies to date have provided a significant effect resulting in diminished pathogenic mechanisms or, perhaps more importantly, lack of cognitive improvement, which would prompt larger, more definitive treatment trials.

Alternative approaches to reduce CNS HIV and/or alleviate NCI in PWH have typically used agents that address pathogenic mechanisms of HIVE or HAND. Animal models and in vitro studies have often focused on reducing the inflammatory consequences of HIV infection in the CNS.

PPAR-gamma agonist studies include those in the EcoHIV model, which showed that HIV and induced inflammatory markers could be reduced by a PPAR-gamma agonist [161,166] (Table 2). If safely combined with ART in an appropriate animal model, this could also be investigated in PWH who have NCI.

PD1 inhibition with pembrolizumab shows promise as a means to potentially eliminate HIV-infected microglia and a Phase I trial in humans to determine safety is underway [181]. However, one concern is that CTL induced killing of HIV-infected microglia in brain could result in bystander immune mediated damage causing neurotoxic effects.

Curcumin was tested in a mouse model of HAND and reversed NCI (see results above including Figure 2 and Table 2). Although the data do not allow the full analysis of pathology associated with NCI and the period of treatment is short, curcumin is inexpensive and tolerated in the human population. Therefore, it may be worth consideration testing in PWH who have NCI.

As discussed earlier, intranasal insulin treatment has been studied in a feline model of retrovirus brain infection and in the EcoHIV mouse model [159,191] (Table 2). Treatment in the FIV model had both antiviral and anti-inflammatory effects [191]. Importantly, insulin improved cognitive performance and, in addition, preserved neuronal integrity and prevented neuronal loss. In the EcoHIV mouse model, intranasal insulin ameliorated cognitive dysfunction and dendritic injury [159]. This prompted a small placebo controlled trial of intranasal insulin in PWH with NCI [262]. Briefly, 21 PWH and mild to moderate NCI were randomized to receive intranasal insulin or placebo. A statistically significant effect on cognitive testing was found. There are plans to pursue a larger clinical trial of this promising treatment.

The data for Jak inhibitors that were tested in vitro, in an animal model of HAND and for safety in PWH are compelling [111,116,120,155]. Our group will be evaluating baricitinib for HIV CNS persistence in a Phase II double blind placebo-controlled trial. Briefly, in our murine model of HAND, we showed that baricitinib crosses the blood brain barrier (BBB), ameliorates phenotypic markers of HAND, and decreases HIV CNS persistence in the brain. Our group has also published evidence that baricitinib efficiently cross the BBB in the rhesus macaque model [263]. Specifically, baricitinib was administered orally to rhesus macaques and then tissues were studied post-mortem, including CSF, lung, and brain. Average CSF baricitinib concentrations were >10% that of plasma (0.29 nanograms(ng)/mL compared to 2.13 ng/mL). CSF concentrations were stable 24 h after dosing, suggesting a large therapeutic window. Brain baricitinib concentration was almost 50% of lung concentration (2.09 ng/g versus 4.43 ng/g) [263]. For our upcoming trial in humans, baricitinib will be compared to placebo with respect to effects on CNS virus including cell associated RNA and DNA as well as inflammation, neuroimaging, and cognition.

## 5. Conclusions

HIV associated NCI remains a significant concern in many PWH, even when HIV is controlled with ART. A significant proportion of individuals with NCI will progress towards dementia, which leads to substantially reduced quality of life, shortened life span, and is costly. Given that many are surviving into middle and older age, comorbidities, particularly those that are common to older populations (e.g., AD), may further contribute to cognitive decline and onset of dementia. More effective CNS control of HIV or better elimination of brain virus through ARV development, plus adjunction treatment of pathogenic consequences of HIV brain infection is needed. Optimal treatment strategies might also address pathogenic mechanisms that are shared by common comorbidities that adversely affect the brain. Clinical studies, animal models, and in vitro data are reviewed with special attention to novel treatment strategies that are supported by the confluence of these data. The hope is that ongoing and future clinical trials investigating these promising approaches will lead to improved quality of life for many PWH.

## Figures and Tables

**Figure 1 microorganisms-10-02244-f001:**
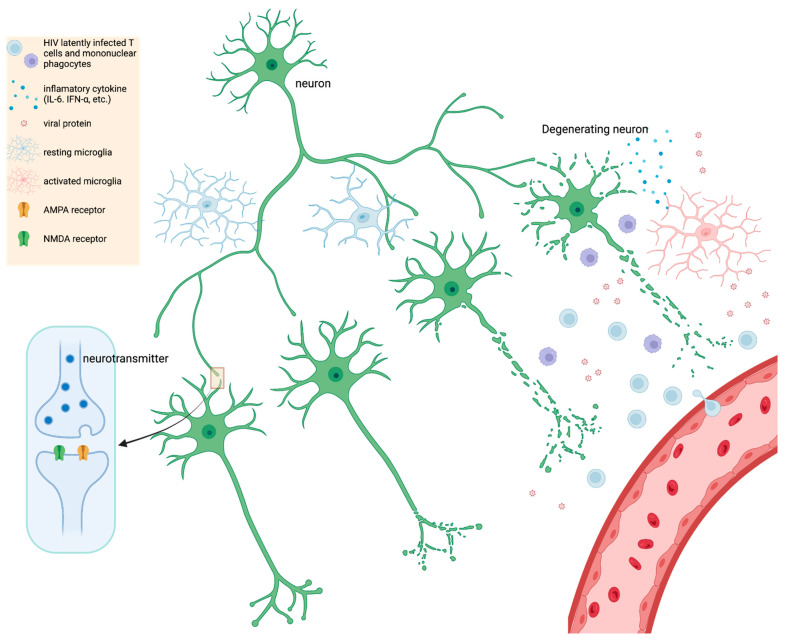
A simple schematic illustrating HIV infection within the CNS. HIV latently infects T cells and mononuclear phagocytes that migrate across the BBB; various stimuli trigger reactivation of viral latency in CNS. Upon activation, elevated levels of viral proteins (gp120, Nef, Tat) trigger resting microglia, which further leads to proinflammatory cytokine secretion, such as IL-6, IFN-α, etc. Chronic inflammation and HIV protein expression contribute to neurocognitive impairment (NCI) and injury of synapses and dendrites (inset diagram). Synaptodendritic injury may include less branching of dendrites, shortening of neurites (axons and dendrites), or downregulation of glutamate receptors such as NMDA or AMPA. In severe but rarer cases of HIV-induced inflammation and NCI, death of neurons can occur. Figure created in BioRender.

**Figure 2 microorganisms-10-02244-f002:**
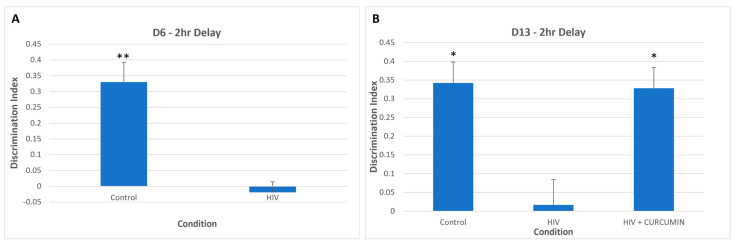
Object recognition testing of Control, HAND (HIV) saline treated, and HAND (HIV) curcumin treated (50 mg/kg) mice. Day 6 with 2 h delay, before introduction of the novel object, in Control and HAND mice (**A**). Day 13 with 2 h delay in Control, HAND and HAND treated with curcumin (**B**). * *p* =0.00015,** *p* = 0.000003.

**Table 1 microorganisms-10-02244-t001:** Summary of Selected Clinical Trials in Review.

Therapeutic	Anti-Inflammatory in CNS	AntiHIV in CNS	Other Effects	Randomized-Controlled?	Group *n* (Analyzed) and Summary Stats *	Outcome(s)
* **Antiretroviral** *						
Maraviroc	No decrease in CSF neopterin and β2-microglobulin. Ref [92]	NM	High EC_50_ in microglia in vitro. Ref [91]	✓	Maraviroc arm n = 9 Control arm n = 5 Time-treatment arm interaction: (*p* < 0.05) Effect size: *(d* = 0.77) and (*d* = 0.55) after 6 months or 12 months, respectively.	Improved Cognition in PWH based on time-treatment arm interaction. Ref [92]
Cenicriviroc	NM	NM	N/A	Open label	Single arm (Cenicriviroc) n = 17 Cognitive domain of attention (*p* = 0.011) and working memory (*p* = 0.017). Decreased plasma levels of sCD163, sCD14 and neopterin (*p* < 0.01).	Improved Cognition; decrease in myeloid activation markers. Ref [94]
Dolutegravir + Maraviroc	NM	NM	Increased CD4+ and CD8+ cell counts.	✓	Dual placebo n = 63 Dolutegravir and placebo n = 67 Dolutegravir and Maraviroc n = 60 cognitive testing (*p* > 0.10).	No improvement in cognition in PWH + NCI. Ref [96]
* **Misc. drugs** *						
Lithium	NM	NM	Lithium is well tolerated with ART.	✓	Placebo arm n = 31 Lithium arm n = 30 Summary Global Deficit Score-24 Weeks (*p* = 0.329)	No improvement in cognition in PWH + NCI vs. placebo arm. Ref [97]
Selegiline transdermal system (STS)	NM	NM	Improvement in psychomotor speed in two prior pilot studies. See reference [98].	✓	3 mg/24 h STS (n = 42) 6 mg/24 h STS (n = 43) Placebo (n = 43) NPZ-8 score *p* = 0.35. NPZ total score *p* = 0.88. (Oxidative stress) CSF protein carbonyl concentration (*p* = 0.260)	No improvement in cognition in STS vs placebo arm. No effect on oxidative stress. Ref [98,99]
Paroxetine, Fluconazole, or both	NM	NM	N/A	✓	Paroxetine + Fluconazole n = 11 Paroxetine + Placebo n = 11 Placebo + Fluconazole n = 9 Placebo + Placebo n = 10 Paroxetine arms improved summary NPZ-8 score (*p* = 0.023)	Fluconazole did not have an observable additive effect with Paroxetine. Paroxetine in general improved cognition in PWH. Ref [100]
Atorvastatin	X	X	Reduction of blood lipids.	Open Label	Atorvastatin (no ART) single arm n = 7 viral load and CSF inflammatory markers (*p* > 0.05) Reduced cholesterol and LDL at 4–8 weeks (*p* < 0.01) or triglycerides at 4 weeks (*p* < 0.05).	No effect on viral load or inflammatory markers in CSF. Ref [106]
Memantine	NM	NM	Significant increase in NAA/Cr in FWM (*p*= 0.040) and Parietal Cortex (*p*= 0.023) via MRS.	✓	Memantine arm n = 54 Placebo arm n = 56 NPZ-8 score *p* = 0.585 (week 16)	No improvement in cognition in PWH. Ref [110]
* **Anti-inflammatory** *						
Minocycline	NM	NM	Decrease in oxidative stress markers after 24 weeks. Ref [108] Decrease in various oxidative stress lipid markers (*p* ≤ 0.024). Ref [108] No effect on various markers of inflammation (*p* > 0.05) Ref [108].	✓	Minocycline arm n = 26 Placebo arm n = 26 Cognitive analysis by UNP Sum criteria (*p* = 0.37) Ref [109]. Minocycline arm n = 8 Ref [108] Placebo arm n = 13 Ref [108]	No improvement in cognition in PWH. Ref [109]

Abbreviations: reference (Ref), central nervous system (CNS), people with HIV (PWH), neurocognitive impairment (NCI), antiretroviral therapy (ART), magnetic resonance spectroscopy (MRS), cerebrospinal fluid (CSF), N-acetyl aspartate to creatine ratio (NAA/Cr), frontal white matter (FWM), neuropsychological z score (NPZ), (NPZ-8) average z score of 8 neuropsychological tests, Uganda Neuropsychological Test Battery Summary Measure (UNP Sum). NM: not measured in referenced study or discussed in review. N/A: not applicable or no further comment. X = No. ✓ = Yes. * *n’s* reported for all studies only include patients who completed the clinical trial.

**Table 2 microorganisms-10-02244-t002:** Selected treatments in animal models and in vitro systems.

Treatment	System	Anti-Inflammatory Properties?	Anti-Retroviral Properties?	Other Effects	Recommended for Future Clinical Trials?
* **Antiretroviral** *					
Nanodiamond carrier + efavirenz	Human neuroblastoma cells; primary human macrophages	NM	✓	Prolonged retention in HIV-infected macrophages. Non-toxic to neuroblastoma cells. Ref [250]	Needs further study in animal models.
PLGA encapsulated elvitegravir	In vitro human BBB model, primary human MDMs; SCID mouse model.	NM	✓	Enhanced HIV suppression in MDMs and mouse model. Pronounced penetration in BBB model in vitro. Ref [252,253]	Needs further study in animal models with larger groups.
Nano-NRTIs	Human primary MDMs injected intracerebrally in SCID mice; primary rat neuron culture.	✓	✓	Less toxic than free NRTIs. Ref [256]	Various nano delivery systems (like the two formulations above) are a potential avenue of research.
* **Miscellaneous treatments** *					
JHU083	EcoHIV Mouse Model	NM	NM	Neuroprotectant and reverses cognitive dysfunction in infected mice. Ref [160]	✓
Didehydro-Cortistatin A (dCA)	BLT mice Ref [173]; Tat Transgenic mice + glia cell lines. Ref [174]	In glia cell line. NM in BLT model.	✓	HIV Tat inhibitor. Reverses dopaminergic system dysfunction linked to Tat neurotoxicity. Ref [173]	Further studies examining cognitive impairment in vivo warranted.
Natalizumab	SIV Model	NM	✓	Neuroprotective. Inhibited trafficking of SIV infected monocytes to CNS and gut. Ref [207]	Further animal studies that investigate blockage of infected monocyte trafficking is worth consideration.
anti-JAM-A or anti-ALCAM antibodies	in vitro human BBB model	NM	✓	Blocks transmigration of infected monocytes across BBB model. Ref [248]	Warrants testing in animal models.
* **Anti-inflammatory** *					
B18R	Human primary MDMs injected intracerebrally in SCID mice.	✓	NM	Treatment reversed cognitive deficits in HIV infected mice. Ref [137]	✓
Curcumin	Human primary MDMs injected intracerebrally in SCID mice.	Reduction in mouse myeloid activation markers (trend).	X	Treatment reversed cognitive deficits in HIV infected mice. Ref [154]	Further testing in HIV + NCI animal models warranted.
Ruxolitinib	Human primary MDMs injected intracerebrally in SCID mice.	✓	✓	N/A Ref [155]	See Ref [111]. Authors did not examine cognition.
Baricitinib	Human primary MDMs injected intracerebrally in SCID mice.	✓	✓	Treatment reversed cognitive impairment in HIV infected mice. Ref [120]	✓
Intranasal Insulin	EcoHIV Mouse Model	✓	✓	Treatment reversed cognitive impairment in HIV infected mice. Ref [159]	✓
Intranasal Insulin	FIV Model	✓	✓	Treatment improved neurobehavioral performance in FIV infected cats. Ref [191]	✓
PPARγ agonists	EcoHIV Mouse Model; gp120 neurotoxicity mouse model	✓	✓	See Ref [161,166,167].	✓
Immunophilin Ligand FK506	Gp120 transgenic mouse model	✓	NM	Mitigates mitochondrial dysfunction and synaptodendritic injury. Ref [175]	✓
Dimethyl fumarate (DMF)	SIV model; primary rat cortex neurons + conditioned media.	In vitro. Ref [221,222]	In vitro. Ref [221,222]	Neither anti-inflammatory nor anti-SIV in SIV model. Ref [223]	Future studies combining DMF with ART in animal model worth considering.
IFN-β	gp120 transgenic mouse model; primary rat and mouse neurons.	✓	NM	Reduces synaptodendritic injury in vivo. Ref [83]	✓
MCC950	gp120 transgenic mouse model; primary mouse neurons and microglia.	✓	NM	Reduces synaptodendritic injury in vivo. Ref [232]	✓
PERK inhibitor GSK2606414	Mouse microglia cell line.	✓	NM	N/A Ref [237]	Warrants further studies in animal models.
Rapamycin (autophagy inducer)	Primary human fetal neurons and human astrocytes.	NM	NM	Neuroprotective in vitro. Ref [246]	Autophagy inducers warrants further studies in animal models.

Abbreviations: Reference (Ref), monocyte-derived macrophages (MDMs), central nervous system (CNS), people with HIV (PWH), neurocognitive impairment (NCI), cerebrospinal fluid (CSF), bone-liver-thymus (BLT), feline immunodeficiency virus (FIV), simian immunodeficiency virus (SIV), interferon-beta (IFN-β), protein kinase RNA-like ER kinase (PERK), blood-brain barrier (BBB), poly (D, L-lactide-*co*-glycolide) (PLGA), Nucleoside reverse transcriptase inhibitors (NRTIs). NM: not measured in referenced study or discussed in review. N/A: not applicable or no further comment. X = No. ✓ = Yes.

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
