# Peer review of "A Rationale and Approach to the Development of Specific Treatments for HIV Associated Neurocognitive Impairment"

_microorganisms, 2022, doi:10.3390/microorganisms10112244_

Round 1
Reviewer 1 Report
The manuscript entitled: “A Rationale and Approach to the Development of Specific Treatments for HIV Associated Neurocognitive Impairment” by Scanlan et al. is a comprehensive review on HIV Associated Neurocognitive Impairment (NCI). The authors discuss extensively the recently developed therapeutics approaches for HIV NCI. The review is well written, and all different aspects are discussed properly. However, I would ask for minor revisions to further improve it:
Line 16: As the HIV field continues to investigate cure strategies, adjunctive therapies to ART are greatly needed. …cure strategies “and” adjunctive therapies to ART are greatly needed.
Line 59: Please define: ADLs
Following “page 5”, where “Table 1” starts is not numbered. Then the page numbering starts again at 2-6 (ending of Table 1). Followed by an unnumbered page and the numbering continues again with “page 2”. It’s not clear if this is only a formatting error and if the page order is correct. However, if this is indeed the correct order, I would suggest that “Figure 1” along with the “2.6 Pathology” section could go before “2.5. Treatment” and “Table 2”.
I would suggest that authors shorten the subsections presented in “2. Clinical and Neuropathological Features of HIV Associated Cognitive Dysfunction”, especially the “2.2. Imaging” subsection. These aspects are more extensively discussed in the available published reviews. Shortening that part of the review will highlight the sections discussing the recent developments and future approaches for HIV NCI treatment, which is the more innovative aspect of this review.
Reviewer 2 Report
This manuscript reviews the current data associated with the development of HIV associated neurocognitive impairment and lays out a number of approaches that can be studied too improve the cognition of persons with mild neurocognitive disorder (MND). As the authors note, it is impossible to cover the whole field and thus, by necessity some areas are left out or given minimal attention. Overall, however, the manuscript is well-written and has given thoughtful suggestions to approaches that could be used to potentially improve those persons with HIV and MND.
Specific comments that should be considered
1. The limitations of animal models should be discussed. There is little evidence that any of the animal models have provided reliable insights into the treatment of HIV-related neurocognitive disorders and only through clinical trials in persons living with HIV (PLWH) potential therapies be identified.
2. Similarly, limitations of in vitro studies using continuous cells lines and non-human cells should be discussed. In vitro studies must be performed in primary human cells to provide useful information on the interactions between HIV in humans.
3. In several places the authors note that monocytes carrying integrated HIV preferentially migrate across the blood-brain-barrier… (ex. page 10, line 674). Please identify the evidence for this preferential migration.
4. The authors should note that the CNS is seeded early in infection (during primary infection). Although the CNS may continue to be seeded, an abundance of evidence indicates infection occurs during primary HIV infection (Consider PMC3490695).
5. In the discussion of HIV proteins, it is important to note that during latency when no infectious virus is being made that viral proteins are still being released by infected cells including gp120, and Tat which can result in neurotoxicity. Additionally HIV ssRNA by itself can result in inflammation through the activation of the NLRP3 inflammasome (consider PMCID: PMC6493331).
6. Page 13, lines 818-821. It is not necessarily desirable to have more efavirenz enter the CNS since EFV has demonstrated significant neurotoxicity. It is true that nanoparticles have the potential to improve CNS penetration of drugs (consider PMID: 23889591).
7. page 12, lines 786-797) HIV Nef has many actions. An important action with regards to Nef is the binding of GAPR-1 and binding to Beclin-1. A Beclin-1 peptide induces autophagy (consider PMC3788641).
Reviewer 3 Report
Line 17 - Extra spacing at the start of the line “HIV imaging…”
Line 18 - missing Oxford comma after “continual inflammation”
Line 21 - missing the in front of “HIV replication”
Line 43 - extra “is” in this sentence “in the Future Directions for the HIV NCI treatment space is we make”
Line 59 - Define what is ADL “ significant impairment of ADLs”
Line 131 - Should the acronym be PWH instead ? “One longitudinal study found that PWHIV on a stable ART”
Line 133 - Should be “are” “reduction of Glx and NAA in the specific regions is associated with NCI”
Line 150 - missing connector words “which is” in this sentence “ translocator protein (TSPO) highly expressed by activated microglial cells”
Line 154 - extra space in this line between astrocytes and during “microglia and astrocytes during neuronal injury”
Line 157 - extra spacing at the start of the line “Interestingly, one PET study..”
Line173-179 - Check for extra spacing and use of capital vs lower case for start of the numbered points for this sentence, “The authors acknowledge the limitations of their comparative review 173 due to:..”
Line 228 - define EC50 “shown to have higher EC50 ”
Line 237 - sentence structure is awkward “Currently, whether a higher a CPE score can improve cognitive outcomes is debatable. “
Line 285 - extra space at start of the line “The two treatments together were ..”
Line 320 - extra space at start of the line, “Downregulating this key cell survival marker..”
Line 336 - extra space at start of the line, “These data underscore the delicate interplay..”
Line 352 - extra space at end of the line , “NCI [124,128,129] .”
Line 357 - extra space at start of line, “A neuronal synapse is composed of”
Line 431 - define SCID, “SCID mice..”
Line 456 - why does the figure 2A not have a column for HIV + Curcumin at day6 ? Curcumin is spelled wrong in the legend for Day 13.
Line 456 - extra space at the start of the line, “In the same preliminary..”
Line 484 - extra space at start of the line, “CaMKII and neurogranin are part of..”
Line 496 - extra space at start of the line, “As excitatory neurons compose..”
Line 555 - extra space at start of the line, “While the BLT model is an attractive model..”
Line 557 - extra space at start of the line, “Preliminary data in our lab suggest..”
Line 622 - extra space at start of the line, “Relevant to all important NeuroHIV animal models..”
Line 645 - define SIV, “SIV infected rhesus macaques..”
Line 788 - should it be only HIV-induced instead ? “HIV-1-induced..”
Line 791 - extra space before the line “successfully ameliorated the..”
Line 838 - extra space at the end of the line, “Translation section above) .”
Line 839 - need space between word and references, “ART[64,66-70].”
